# Diagnosis and Management of Porocarcinoma

**DOI:** 10.3390/cancers14215232

**Published:** 2022-10-25

**Authors:** Kodai Miyamoto, Teruki Yanagi, Takuya Maeda, Hideyuki Ujiie

**Affiliations:** Department of Dermatology, Faculty of Medicine and Graduate School of Medicine, Hokkaido University, Sapporo 060-8638, Japan

**Keywords:** porocarcinoma, histopathology, immunohistochemistry, YAP fusion, chemotherapy

## Abstract

**Simple Summary:**

Porocarcinoma (PC) is very rare and is known to arise from the cutaneous intraepidermal ducts of the sweat glands. Diagnosing PC used to be challenging due to the diverse clinical and histopathological findings. The gene alterations in PC have been discovered in recent years. Although the prognosis of patients with metastatic PC is poor, there has been little evidence to support a specific chemotherapy-based regimen due to the rarity of this cancer. New treatments, such as immunological therapies, have recently been reported. This review focuses on the history, pathogenesis, pathological features, diagnosis, and treatment of PC.

**Abstract:**

Eccrine porocarcinoma, also known as porocarcinoma (PC) and malignant eccrine poroma, is very rare and is known to arise from the cutaneous intraepidermal ducts of the sweat glands. Its etiology is not well understood; however, some studies suggest that PC tumors originate from benign eccrine poroma. Recently, several gene alterations have been reported in PC that can reveal mechanisms of the oncogenic process. Since the clinical and histopathological findings of PC are variable, PC is difficult to diagnose precisely, especially when the histology resembles that of cutaneous squamous cell carcinoma or poroma. Immunohistochemical staining with carcinoembryonic antigen and epithelial membrane antigen may help to distinguish PC from other tumors. The standard treatment for local PC is wide local excision. The prognosis of patients with metastatic PC is poor, with mortality rates of approximately 60–70%. The efficacy of radiation and chemotherapy for metastatic PC is limited; however, immunotherapy with pembrolizumab, a programmed cell death protein 1 inhibitor, could be a promising treatment. This review focuses on the history, pathogenesis, pathological features, diagnosis, and treatment of eccrine porocarcinoma.

## 1. Introduction

Malignant cutaneous adnexal tumors (MCATs) are broadly divided into four groups: eccrine, apocrine, mixed, and un-classified. Porocarcinoma (PC) is an uncommon and aggressive skin cancer that originates in the intraepidermal component of the sweat gland apparatus [1]. PC was first reported in 1963 by Pinkus and Mehregan, who referred to it as “epidermotropic eccrine carcinoma” [2]. Mishima and Morioka were the first to coin the term “eccrine porocarcinoma”, in 1969, and they focused on the common histopathologic transformation of the benign form of eccrine poroma to its subsequent malignant counterpart [3]. After that, various names were used, such as eccrine porocarcinoma, malignant eccrine poroma, and malignant hidroacanthoma simplex. In the 2018 World Health Organization (WHO) Classification of Skin Tumors, PC was used more frequently than other common names in the literature [4]. This review focuses on the epidemiology, pathogenesis, pathological features, diagnosis, and treatment of porocarcinoma.

## 2. Epidemiology

Few studies have addressed the epidemiology of PC because PC is extremely rare. Previous studies reported that PC accounts for about 0.005–0.01% of all malignant cutaneous neoplasms, and PC appears to be slightly more prevalent among males than females [5]. Blake et al. reported incidence and survival patterns of PC from 2001 to 2005 in the United States of America (USA) [6]. They showed an incidence rate for men of 0.05 per 100,000 person-years and for women of 0.02 per 100,000 person-years (in 2000, US standard). The total incidence rate was 0.04 per 100,000 person-years. Merilainen et al. examined the epidemiology and incidence rate data for eccrine porocarcinoma from the Finnish Cancer Registry from 2007 to 2017 [7]. The incidence rate was 0.06 per 100,000 person-years for men and 0.04 per 100,000 person-years for women. Gibbs et al. analyzed the incidence rate and trend for cutaneous adnexal tumors in the United States between 2000 and 2018 [8]. The age-adjusted incidence rate for PC was 0.045 per 100,000 person-years. They also mentioned that the incidence rate for PC did not change during the period 2000–2018. A retrospective study in Minnesota between the years 2000 and 2010 showed an incidence of 0.2 per 100,000 person-years for both males and females [9]. The age-standardized incidence using the European Standard Population from the East of England between 2004 and 2014 was 1.3 for men and 2.4 for women, and the overall average incidence rate was 1.9 per 100,000 person-years [10]. The incidence rate of PC is at least 10 times higher in England than in the locations of the other four reports above. Although this information is difficult to confirm, it is possible that the number of diagnosed cases may increase in the future as the disease becomes more recognized and as improvements in histology reduce the number of misdiagnoses. As described above, the incidence rate of PC seems to be approximately 0.02–0.2 per 100,000 person-years. However, the sex ratio varies from report to report. There seems to be no consistent difference between the sexes.

PC occurs predominantly in elderly people. The incidence increases with age, and the disease is reported to affect mostly those in their 70s and 80s. Some studies have reported mean ages at diagnosis that vary from 67 to 76 years, but these ages converge at around 70 years [6,7,11,12]. A few cases of eccrine porocarcinoma in young patients have been reported. Valverde et al. described the case of an 8-year-old girl, who is the youngest case we have identified [13]. Baptista et al. reported the case of PC in a 12-year-old patient who also had XP (xeroderma pigmentosum) complications [14].

Concerning the locations of PC, Robson et al. reported 69 cases of PC and examined the clinicopathological characteristics [5]. They revealed the lower extremities to be the most common site (44%). Other common sites were the trunk (24%), the head (18%), and the upper limbs (11%). Scampa et al. analyzed 563 cases of PC from the Surveillance, Epidemiology, and End Results (SEER) data [15]. In that analysis, the skin of the lower extremities and hips were the most common sites (190 cases: 33.7%), followed by the skin of the head and neck (30.6%) and the skin of the trunk (19.5%). Domere et al. published a report on PC lesions [16]. In their 83 cases, they reported that the most common primary sites were the head and neck (33.6%). A review by Kim et al. supports the results of the Domere group [12]. Kim et al. reported 37 patients with PC in South Korea, and the most common site for primary lesions was the head and neck (29.7%). The second most common site was the trunk (27.0%), and the lower limbs were third (24.3%). Salih et al. performed a meta-analysis of PC [11]. They searched the medical literature and found 104 studies covering 453 PC patients. They recorded the affected sites for all 453 of these patients. The most common locations were the head and neck (39.9%), followed by the lower extremities (33.9%), the upper extremities (8.8%), the back (5.1%), and the chest (4.6%). De Giorgi et al. reported that the affected sites of PC may depend on the sex, based on a study of 52 PC cases (31 male and 21 female). [17] Their study revealed that the most involved site was the head and neck (47%), which was more common in males than in females (34% males vs. 17% females). The lower extremities were the most common site among female cases (17% females vs. 9% males). Excluding Robson’s and Scampa’s reports, the head and neck are the most common site of PC. The second most common site is the lower extremities or trunk, but if the upper extremities are also considered, the extremities are more likely to be affected. Although race was not stated with respect to the site of predilection, there did not appear to be a significant difference between Koreans and other populations. Further research is needed to determine any racial predilections for the most common sites. 

## 3. Pathogenesis

While the etiology of PC is not fully understood, it is thought to possibly arise from the acrosyringium, which is the intraepidermal spiral duct of the eccrine apparatus [18]. PC may occur de novo or may develop from a pre-existing benign tumor. Poromas, which are benign adnexal neoplasms, account for approximately 10% of eccrine or apocrine sweat gland tumors, and it is well-known that poromas may transform into PC [18,19,20]. Shaw et al. reported that all 27 PC patients in their study had co-existing poromas [19]. Puttick et al. also reported that three patients had poromas for many years before those tumors transformed into PC [21]. A clinicopathological study of 69 PC cases revealed that 18% had adjacent features typical of poroma, which was much lower than in previous reports [5].

Some studies reported that exposure to sunlight is a main risk factor for PC tumorigenesis [11]. This is because the most common sites for PC are sun-exposed areas (the head, neck, and extremities). Puttonen et al. found a negative correlation between UV damage and tumor-infiltrating lymphocytes (TILs) [22]. The results suggest that UV has a potentially immunosuppressive effect on PC microenvironments. On the other hand, in Merkel cell polyomavirus-negative Merkel cell carcinoma, UV exposure correlates with increased TILs. [23] Further studies are needed on the relationship between UV exposure and TILs in PC.

Immunosuppression could also be a risk factor for the development of PC. Mahomed et al. reported 5 PC patients with a history of immunosuppressive medications or conditions [24]. Three of the five cases were renal transplant recipients and the other two cases had acquired immune deficiency syndrome (AIDS). Although the relationship between exposure to chemical agents and the development of PC is rare, several cases of such a relationship have been published. Ioannidis et al. reported a patient with chronic exposure to benzene glue that resulted in PC [25]. Helmke et al. presented a case of PC on the lower leg that had arisen on a scar from a poison gas explosion [26].

The mechanisms behind tumorigenesis in PC remain poorly understood, since few PC investigations have been performed and patients with PC are very few. It has been reported that specific oncogenic drivers, signaling pathways, and cell-cycle dysregulation are involved in PC tumorigenesis. Gene alterations of cancer-associated genes (oncogenic drivers and tumor suppressors) such as *TP53* (tumor protein p53), *CDKN2A* (cyclin-dependent kinase inhibitor 2A), *HRAS* (HRas proto-oncogene, GTPase), *EGFR* (epidermal growth factor receptor), and *Rb1* (Retinoblastoma 1) have also been mentioned [27,28,29,30]. The dysregulation of the MAPK (mitogen-activated protein kinase) pathway and the PI3K-AKT (phosphatidylinositol-4,5-bisphosphate 3-kinase, AKT serine/threonine kinase) pathway has been suggested in other studies [28,29,30].

Several studies have analyzed gene mutations in PC through targeted and whole-exome sequencing (WES). Harms et al. analyzed 5 PC tumors using next-generation sequencing [30]. They detected nonsynonymous mutations, including oncogenic activation and tumor suppressor inactivation events. The tumor suppressor mutations mainly included *TP53* (80%) and *RB1* (60%). In the activating mutations, *HRAS* was detected in 40%. Westphal et al. reported the overexpression of the genes *EGFR*, *PAK1*, and *MAP2K1* (mitogen-activated protein kinase kinase 1; also known as MEK1) in PC tumors as assessed by WES, RNA sequencing (RNA-Seq), and comparative genomic hybridization analyses [31]. Based on this result, they treated the patient with anti-EGFR antibody. Denisova et al. performed WES on 14 samples of PC. The two most common coding mutations were *TP53* and *FAT2* (FAT atypical cadherin 2) (mutated in 43% of cases for each mutation), followed by *KMT2D* (lysine methyltransferase 2D) and *CACNA1S* (calcium voltage-gated channel subunit alpha1 S) (mutated in 36% of cases for each mutation) [32]. *TP53*, which is the cell cycle regulator and tumor suppressor gene, has been reported to be the most common site of genetic mutations in PC [33]. The partial inactivation of the p53 pathway is a key event in both tumorigenesis and tumor progression. *RB1* mutations have also been detected in many cases of PC, and these were truncating nonsense or frameshift mutations that were predicted to result in loss of function. 

A fused gene is a hybrid gene formed from two independent genes by translocation, inversion, or deletion. The byproducts of these genes potentially lack regulatory domains, leading to their overexpression. For this reason, the proteins produced by fused genes can lead to oncogenesis. [34,35] Recently, the relationship between gene fusion and PC oncogenesis has become clearer. Sekine et al. reported that highly recurrent YAP1-MAML2 and YAP1-NUTM1 fusions were found in 92 of 104 poroma cases (88.5%) and in 7 of 11 PC cases (63.6%) [36]. *YAP1*, which encodes a paralogous transcriptional activator of TEAD, is negatively regulated by the Hippo signaling pathway. *YAP1* fusions that strongly transactivate the TEA domain family member (TEAD) reporter and help anchorage-independent growth are involved in tumorigenesis [37]. The *YAP1* fusion product expresses the N-terminal portion of YAP1 in the nucleus and lacks the C-terminal portion [36]. PC may have recurrent translocations, including YAP1-MAML2 and YAP1-NUTM1, which are found most in PC cases. Although the number of cases is still small, the oncogenesis of PC has been revealed. 

## 4. Diagnosis

Diagnosing PC is very challenging because this disorder presents a wide variety of clinical and histological findings. The differential diagnoses include squamous cell carcinoma (SCC), poroma, and other adnexal tumors. The diagnosis should be based not only on the clinical findings, but also on the dermoscopic, histopathological, and immunohistochemical findings [38].

### 4.1. Clinical Features

PC typically presents as a firm, single, dome-shaped papule or nodule (Figure 1a). It can appear skin-colored, erythematous, or violaceous, and it can be asymptomatic or itchy and painful [7]. The lesion may develop as an oozing and ulcerating plaque [39]. Robson et al. revealed PC tumor sizes to range from 0.4 to 20 cm, with the median size of invasive cases being 2.0 cm [5]. The preoperative duration of the lesions also varied from 2 weeks to 60 years (mean of 9 years, median of 4 years).

### 4.2. Dermoscopic Features 

A few studies have been published on the dermoscopic features of PC. PC displays thin vascular patterns that resemble the features of poroma; however, PC has more conspicuous and irregular vessels [40]. This atypical vascular pattern is also observed in other malignant skin lesions, such as amelanotic melanoma, SCC, basal cell carcinoma, actinic keratosis, and Bowen’s disease. For the differential diagnosis, important dermoscopic features of PC are whitish globular structures on a light brown background. The combination of atypical vascular patterns and milky-red globules is specific to PC (Figure 1b). 

### 4.3. Histopathology of Hematoxylin and Eosin Staining

The histological features of PC in hematoxylin and eosin (HE) staining are diverse. The histopathological findings in PC tend to be irregular tumors composed of characteristic poromatous basaloid cells that display ductal differentiation, at least in part, and have marked cytologic atypia [5]. Mitotic activation and necrosis along with epidermal hyperkeratosis are also common characteristics. Polygonal tumor cells accompanied by regular squamous or dyskeratotic cells may be seen in clusters that form multiple intraepidermal nests of various sizes (Figure 2) [41]. In a histopathologic study of 69 PC cases, the formation of mature ducts lined with cuboidal epithelial cells (Figure 3) was the most common finding (68%), followed by intracytoplasmic lumina (39%) and comedo necrosis (32%) (Figure 2) [5]. A retrospective study of PC in South Korea showed similar results [12]. Six of the 9 PC cases (66.7%) showed mature duct formation and necrosis (comedo necrosis/diffuse necrosis), which were the most common findings. Squamous differentiation is also a common finding in PC (five cases, 55.6%). Mahomed et al. analyzed the pathological features of 21 cases and found squamous differentiation in every case [24]. Riera-Leal et al. reported pathological findings of 33 cases, with the three most common features being diffuse necrosis (64%), comedo necrosis (45%), and squamous differentiation (42%) [42]. Based on these studies, the observation of duct formation, comedo/diffuse necrosis, and squamous differentiation appear to be significant in HE staining. However, in a retrospective histopathological analysis of 19 cases by Belin et al., seven cases (37%) were not diagnosed correctly, with five of these cases (26%) misdiagnosed as SCC [38]. Although the pathologic features of PC have been reported, PC remains difficult to diagnose based on HE staining alone.

### 4.4. Immunohistochemistry

Immunohistochemical analysis is a useful tool for diagnosing PC. Several immunohistochemical markers have been used to differentiate benign poroid tumors from malignant tumors. Immunohistochemical staining for carcinoembryonic antigen (CEA) and epithelial membrane antigen (EMA) has been frequently used to identify ductal structures. EMA seems more sensitive than CEA in detecting PC tumor cells. Riera-Leal L et al. [42] and Perm et al. [43] performed immunostaining for CEA and EMA, revealing that EMA showed a higher positivity rate than CEA in the analysis of PC tumors. Shiohara et al. reported that all 11 PC samples were positive for EMA staining [44]. However, Beer et al. performed immunostaining on 24 SCC tumors and found that EMA was positive in 96% (23 cases) and CEA in 30% (seven cases) [45]. Thus, while the identification of ductal structures through immunostaining with CEA and EMA can support the diagnosis of PC, these tests also highlight the eccrine ducts of SCC, suggesting that CEA/EMA immunostaining is unable to lead to a definitive diagnosis of PC. 

Various studies reported specific biomarkers for PC. Goto et al. performed CD117 (KIT) staining for PC tumors (22 cases) and SCC tumors (31 cases) [46]. The Goto group found all of the PC tumors to be positive for CD117, while only six of the SCC cases (19%) were positive for CD117, suggesting that CD117 could be a useful marker for differentiating PC from cutaneous SCC. Immunohistochemical staining for cytokeratin 19, c-KIT, and BerEP4 is also useful when differentiating PC from SCC [47]. Goto et al. revealed that cytokeratin 19 was expressed in 13 of 14 PC cases (92.9%) but was positive in only 9 of 22 SCC cases (40.9%). c-KIT was positive in 11 of 14 PC cases (78.6%), compared to only 3 of 22 SCC cases (13.6%). Moreover, BerEP4 expression also differed between PC and SCC (57.1% vs. 9.1%, respectively).

There have been studies on immunostaining based on advances in genetic analyses for PC tumors. Zahn et al. investigated whether the immunohistochemical expressions of p53, Rb, and p16 could distinguish malignant PC tumors from benign poromas [48]. Of 15 the poroma cases, 14 (93%) consistently displayed the strong but diffuse expression of Rb. No case showed the positive expressions of p53 or p16. Reportedly, 7 of 16 PC cases (44%) displayed diffuse or focal Rb expression, eight of 15 PC cases (53%) showed diffuse loss or overexpression of p53, and 6 of 14 PC cases (43%) showed diffuse loss or overexpression of p16. The study by the Zahn group suggested that abnormal positive results among any of these three tumor suppressors (p53, Rb, or p16) could be a specific marker for PC. Furthermore, the immunohistochemistry of testicular nuclear protein (NUT) can screen for *NUTM1* translocations in PC and poroma [49]. A study of six cases of adnexal skin tumors identified *YAP1* gene fusion [34]. The tumor specimens in all cases of poromas (two cases) and PC (three cases) with *YAP-NUTM1* fusion showed positive immunostaining for NUT. Snow et al. assessed *NUTM1* gene fusions using 13 immunohistochemically NUT-positive poroid tumors [35]. *NUTM1* fusion was detected in 12 of 13 cases through deep sequencing, and *YAP1–NUTM1* fusion was identified in 11 of 12 cases. In 9 PC cases, *YAP-NUTM1* fusions were identified in 7 cases.

## 5. Prognosis and Risk Factors for Metastasis/Recurrence/Disease-Specific Survival

The prognosis for PC seemed to be generally good in early-stage cases, which are almost always treated with surgical resection. The staging for MCATs has been defined by the AJCC TNM staging manual, which is the same staging system for SCC [50] In patients with MCATs, the overall 5-year survival rate was 96% for those with stage I or II. There has been no consensus on the surgical margin for wide local resection (WLE). In general, the average margin is 2–3 cm for surgical resection [51]. WLE has been found to have a local recurrence rate of 20%, even when clear margins are confirmed at primary resection. Therefore, Mohs micrographic surgery (MMS) has been increasingly used, with reports of less local recurrence than with WLE.

Histopathological analysis is important when determining tumor invasion and for the prediction of metastasis/recurrence. Robert A et al. reported that the tumor borders of PC can be divided into “infiltrative”, “pagetoid”, or “pushing” [5]. “Infiltrative” is characterized by an ill-defined lower margin of malignant clusters infiltrating the dermis. “Pagetoid” is identified by the intraepidermal spread of tumor cells mimicking Paget’s disease. “Pushing” PC lesions often have polypoid tumors with distinct dermal limits [38]. Robert A et al. also reported that some histopathological characteristics reflect poor prognosis, known as “poor-prognosis histopathological signs” [5]. The signs are the presence of lympho-vascular invasion, more than 14 mitoses per high-power field, and tumoral involvement deeper than 7 mm. [14] Koh et al. investigated epidemiological and pathological features of 52 PC cases and determined several high-risk features [52]. Tumor depth was associated with both increased age and clear cell differentiation, suggesting it as a high-risk factor with poor prognosis.

As for the recurrence rates, a retrospective study revealed that 6 of 52 cases (11.5%) showed recurrence of PC tumors within 5 years [17]. Since the reported recurrence rates vary from study to study, Belin et al. proposed an algorithm using prognostic histological factors that could guide surgical procedures [38]. First, they classified biopsy specimens into the three subtypes of infiltrative, pushing, and pagetoid as described above. If the resected sample was pushing PC, no additional resection was performed, but if it was infiltrative or pagetoid, they recommended additional MMS.

Scampa et al. reported the 5-year overall survival (OS) as 74.8% based on an analysis of 563 PC cases [15]. They also mentioned that the prognosis for PC varies greatly between early and advanced stages. In patients with advanced/metastatic PC, the prognosis is poor. A 2017 meta-analysis of 453 cases reported that 110 cases (31%) presented metastasis, and the most common metastatic organ was the nearby lymph node (58.5%), followed by the lungs (12.8%) [11]. Song et al. mentioned that in the presence of lymph node metastasis, the mortality rate is as high as 67%, and the probability of distant metastasis is about 11–12% [51]. Robson et al. found that patients with lymph node metastases had 1-year and 3-year overall survival (OS) of 88.9% and 39.5%, respectively [5]. No studies have directly compared OS for metastatic SCC to that for metastatic PC. Hillen et al. analyzed 190 cases of advanced SCC [53] and found a 1-year OS of 84% and a 3-year OS of 47% for metastatic SCC. Based on these results, the overall survival rate for metastatic PC may be worse than that for metastatic SCC at 3 years. Concerning the prognostic difference between SCC and PC, further studies are needed.

## 6. Treatments

### 6.1. Surgical Treatments

Surgical resection is almost always performed in the early stages of the disease or in resectable cases. WLE is the main treatment for localized PC tumors. Treatment using MMS has also been increasingly reported in recent years. A 2020 meta-analysis of 120 PC cases revealed that 92.5% of patients were treated with surgical resection [54]. Of the surgical resection cases, WLE and MMS were performed in 76.7% and 15.8% of cases, respectively.

Few reports have addressed sentinel lymph node biopsy (SLNB) in MCATs, and no definite conclusions have been reached. Storino et al. investigated 41 patients diagnosed with MCATs [50]. Of the 25 patients who underwent SLNB, only one had lymph node metastasis, suggesting that SLNB may not be necessary in early-stage cases of MCATs. However, if the SLNB was positive, metastatic lesions could be detected early and systemic treatment could be initiated. SLNB could be considered depending on the patient’s condition and the presence of advanced disease risk factors.

### 6.2. Radiotherapy

There are no clear criteria for radiation therapy (RT) for PC [55]. There has been only one case in which primary RT was used to treat PC lesions that were not suitable for surgery [54]. Reports on the use of adjuvant RT after surgery are also limited. Adjuvant RT has generally been considered helpful for lymph node metastases, perineural invasion, positive resection margins, high-grade tissue, multiple lesions, and recurrent disease. It is often used in combination with chemotherapy [56]. Fionda et al. published a systematic review of adjuvant RT against PC, [55]. suggesting that postoperative adjuvant RT may be effective in cases with positive or close margins and unfavorable histology.

### 6.3. Chemotherapy

There has been little evidence for a specific chemotherapy-based regimen in metastatic PC due to the rarity of this cancer. We therefore have summarized cases in which chemotherapies have been conducted and reported in detail (Table 1).

A total of 60% of patients with advanced PC were mainly treated with platinum-containing drugs such as carboplatin (CBDCA) and cisplatin (CDDP). Fluorouracil (5-FU) is often used in combination with CDDP. McGuire et al. reported a case of PC on the right thumb. [60] After amputation of the thumb, metastatic PC was identified in the right chest wall, the right anterior axilla, and the right radial wrist. Chemotherapy with CBDCA, paclitaxel, and IL-2 injections, as well as radiotherapy to the right axilla, was effective against the metastatic lesions (Case #4 in Table 1). The patient has been followed up for one year without recurrence. Chow et al. reported a case of PC with lymph node metastasis (Case #8) [64]. Chemotherapy with CDDP and docetaxel, followed by radiotherapy, eliminated the tumors completely, and the patient has been in remission for 18 months. Based on our review of the previously reported cases, only 31.3% of cases treated with CBDCA or CDDP showed a therapeutic response (CR or PR), suggesting that PC is relatively resistant to cytotoxic agents. Some groups have reported targeted therapies for metastatic PC, especially cetuximab [65,75]. Cetuximab is a monoclonal antibody that targets the epidermal growth factor receptor (EGFR) [80]. Godillot et al. reported a case of PC in which cetuximab was observed to be effective (Case #9) [65]. Since they confirmed the protein expression of EGFR on the PC tumor, they chose cetuximab and paclitaxel for the treatment of patients with distant metastases to the lymph nodes, lungs, and bones. The tumors disappeared completely and have not recurred for 6 months.

Immunotherapy with pembrolizumab, which is a programmed cell death protein 1 (PD-1) inhibitor, could be a novel effective treatment for metastatic PC. Although cases are few (three cases), two cases with metastatic PC achieved good courses [58,67,75]. Lee et al. reported the first PC case treated with pembrolizumab (Case #2) [58]. Their case received radiotherapy and chemotherapy (12 cycles of carboplatin and capecitabine) for distant metastases to the brain, but the disease continued to progress. After a switch to pembrolizumab, the tumor showed obvious regression and disappeared after 16 months. The tumor has been in remission for 22 months. A case reported by Singh et al. has remained in partial remission for 18 months from the administration of pembrolizumab (Case #11) [67]. As seen above, pembrolizumab may have the potential to become an alternative treatment, or even a primary treatment, for metastatic PC. However, further studies are needed to establish its efficacy and safety in PC patients.

## 7. Conclusions

The exact incidence rate of PC is not known, but based on past reports, it appears to be about 0.02 to 0.2 per 100,000 person-years. The number of patients with PC is likely to increase because of its predilection for the elderly. The diagnosis of PC used to be challenging due to the diverse clinical and histopathological findings. The gene alterations in PC have been discovered in recent years, and it is possible that they could be linked to diagnosis and treatment. Combined with advances in immunostaining, we can expect more accurate diagnosis in the future. The prognosis for early-stage PC is basically good with only surgical resection, but the recurrence rate is about 20%. Chemotherapy-based regimens for recurrent and metastatic cases have not been established. Immunotherapies, such as of pembrolizumab, could be effective treatments for metastatic PC. However, because of the small number of cases using pembrolizumab, further studies are needed to prove the efficacy and safety of this treatment.

## Figures and Tables

**Figure 1 cancers-14-05232-f001:**
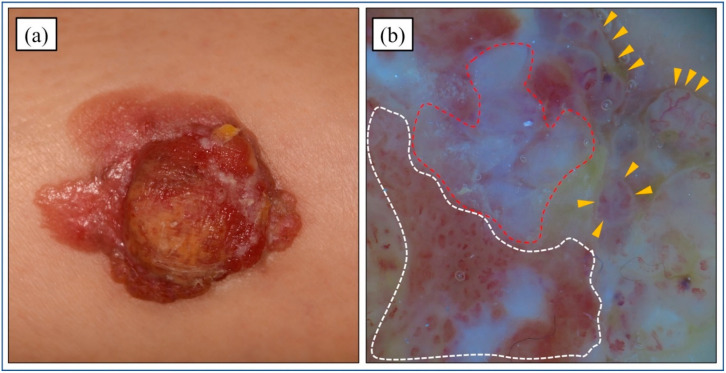
Clinical and dermoscopic findings. (**a**) A brown dome-shaped nodule of 30 × 35 mm is observed on the right thigh. The erythema around its margins is asymmetric. (**b**) Dermoscopic examination shows whitish globular structures (red dotted line). Polymorphous vessels with a whitish negative network on a light brown background are also observed (yellow arrowheads). The vessels are more conspicuous and irregularly shaped (white dotted line).

**Figure 2 cancers-14-05232-f002:**
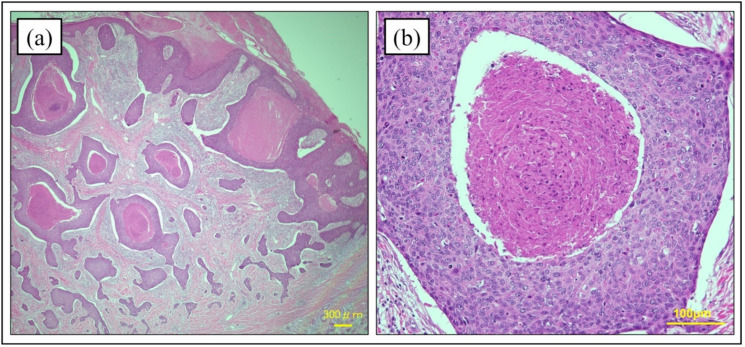
Histopathological findings of comedo necrosis. (**a**,**b**) Clusters form irregular nests of various sizes in the intraepidermal. The large nests are accompanied by many comedo necroses. (Hematoxylin and eosin stain; (**a**) original magnification ×20, (**b**) original magnification ×200).

**Figure 3 cancers-14-05232-f003:**
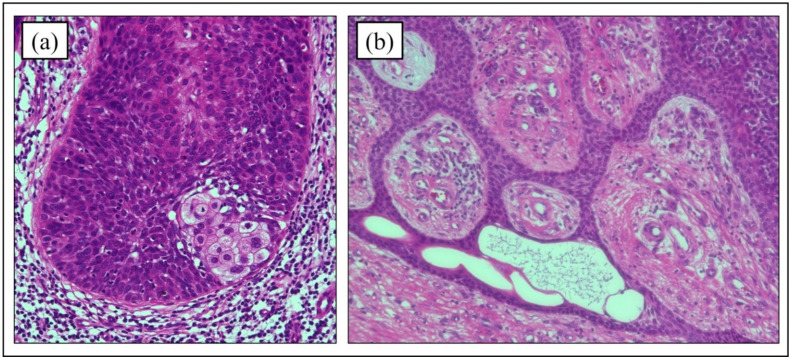
Histopathological findings of ductal formation. (**a**) Clear cell changes in the tumor are observed. (**b**) Mature ductal formation and atypical cell proliferation are found. (Hematoxylin and eosin stain; (**a**) original magnification ×200, (**b**) original magnification ×200).

**Table 1 cancers-14-05232-t001:** A summary of 28 cases treated with systemic chemotherapies.

Case	Clinical Findings						Treatment			Clinical Course		
	Age	Sex	Site	Size (mm)	Lymph Node Metastasis	Distant Metastasis	Surgical Resection	Chemotherapy	Radiation Therapy	RECIST	Postoperative Metastasis	Reference
1	42	M	axillary	n.d.	+	-	+	bleomycin + electric pulse	-	CR	-	[57]
2	67	F	lower limb	n.d.	-	-	+	CBDCA + CAPE, Pembrolizumab	+	CR	+	[58]
3	77	M	foot	n.d.	+	-	+	CBDCA + MMC	-	CR	-	[59]
4	56	M	thumb	18 × 13	+	-	+	CBDCA + PTX + IL-2	-	CR	+	[60]
5	61	M	leg	30	+	-	+	CDDP + 5-FU	+(50.4 Gy/28 Fr)	CR	+	[61]
6	67	M	foot	n.d.	+	-	+	CDDP + 5-FU	+(50 Gy/25 Fr)	CR	+	[62]
7	54	M	breast	n.d.	-	-	+	CDDP + 5-FU, DTX	-	CR	+	[63]
8	63	M	palm	50 × 60	-	-	+	CDDP + DTX	+(50 Gy/25 Fr)	CR	+	[64]
9	64	F	neck	50	+	-	+	PTX + cetuximab	+(57.5 Gy)	CR	+	[65]
10	71	M	thigh	40 × 35	-	-	+	VCR	+(45 Gy)	CR	+	[66]
11	79	M	scalp	n.d.	+	-	+	pembrolizmab	+	PR	-	[67]
12	67	M	neck	30 × 25	-	-	+	PTX + IFN-α	-	PR	+	[68]
13	75	M	leg	91	+	-	+	UFT	-	SD	-	[44]
14	72	M	thigh	17 × 12	-	-	+	CBDCA + farmourubicin	-	PD	+	[69]
15	67	M	scalp	30 × 20	-	-	+	CBDCA + DTX	+	PD	+	[70]
16	81	F	buttock	25	-	-	+	CDDP+ 5-FU	-	PD	+	[44]
17	62	F	head	98	+	-	+	CDDP + ADM + VDS, PEP + 5-FU, CPA + pirabubicin, ACNU + UFT	+(50 Gy)	PD	+	[44]
18	54	F	vulva	10 × 6	-	-	+	CDDP, PTX + CBDCA	+(50.4 Gy/28 Fr)	PD	+	[71]
19	44	M	scrotum	60 × 70	+	+	-	CDDP + 5-FU	-	PD	N/A	[72]
20	62	M	scalp	5 × 6	+	+	-	CDDP + 5-FU	-	PD	N/A	[73]
21	50	M	arm	80	+	+	+	CDDP + DTX	+(21 Gy)	PD	+	[74]
22	70	M	scalp	n.d.	+	-	+	DTX, pembrolizumab, cetuximab + CAPE	-	PD	+	[75]
23	79	F	leg	n.d.	-	-	+	IFN	-	PD	+	[76]
24	64	M	leg	16	-	-	-	MMC + VCR + EPI-ADM + CDDP + 5-FU + PEP, CDDP + 5-FU	+(50 Gy)	PD	N/A	[44]
25	67	F	neck	n.d.	+	-	+	platinum derivatives + taxanes	+(50 Gy/25 Fr)	PD	+	[77]
26	77	F	leg	n.d.	-	-	+	PTX + CAPE + IFN-α	-	PD	+	[76]
27	64	M	lateral wall	n.d.	+	+	-	CBDCA + taxanes	+(53 Gy)	n.d.	N/A	[78]
28	67	M	hip	25 × 25	+	-	+	etoposide, VDS, CBDCA	-	n.d.	-	[79]

n.d., not described; N/A, not applicable; RECIST, Response Evaluation Criteria in Solid Tumors; CBDCA, carboplatin; PTX, paclitaxel; DTX, docetaxel; 5-FU, fluorouracil; ACNU, nimustine; ADM, doxorubicin; CDDP, cisplatin; CPA, cyclophosphamide; EPI-ADM, epirubicine; MMC, mitomycin C; PEP, pepleomycin; UFT, tegafur-uracil; VCR, vincristine; VDS, vindesine; CAPE, capecitabine.

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
