# Peer review of "Diagnosis and Management of Porocarcinoma"

_cancers, 2022, doi:10.3390/cancers14215232_

Round 1

Reviewer 1 Report

The prognosis being significantly worse than other cutaneous epithelial neoplasms remains enigmatic.  Because the number of cases is so small, it is possible that it may not be as poor as other neoplasms when matched thickness for thickness, the degree of atypia, anaplasia and number of mitoses, etc.  Furthermore, it is noted that patients who develop the neoplasm may be immunocompromised.  Is the prognosis truly worse than a large, deeply invasive poorly differentiated squamous cell carcinoma or pleomorphic dermal sarcoma?  It would be valuable to include a discussion that encompasses these points.

Author Response

Reply to Reviewer #1:

The prognosis being significantly worse than other cutaneous epithelial neoplasms remains enigmatic.  Because the number of cases is so small, it is possible that it may not be as poor as other neoplasms when matched thickness for thickness, the degree of atypia, anaplasia and number of mitoses, etc.  Furthermore, it is noted that patients who develop the neoplasm may be immunocompromised.  Is the prognosis truly worse than a large, deeply invasive poorly differentiated squamous cell carcinoma or pleomorphic dermal sarcoma?  It would be valuable to include a discussion that encompasses these points.

We would like to thank Reviewer #1 for the insightful comments. As was pointed out, we have added the prognostic data of cutaneous SCC and porocarcinoma. No studies have directly compared OS for metastatic SCC versus metastatic PC. Hillen et al. analyzed 190 cases of advanced SCC and found a 1-year OS of 84% and a 3-year OS of 47% for metastatic SCC. Compared to metastatic SCC, the overall survival rate for metastatic PC may be worse at 3 years; however, further studies are needed. We have added this information to the text as follows (5. Prognosis and risk factors for metastasis/recurrence/disease specific survival):

Page 8, lines 309–314

No studies have directly compared OS for metastatic SCC to that for metastatic PC. Hillen et al. analyzed 190 cases of advanced SCC[53] and found a 1-year OS of 84% and a 3-year OS of 47% for metastatic SCC. Based on these results, the overall survival rate for metastatic PC may be worse than that for metastatic SCC at 3 years. Concerning the prognostic difference between SCC and PC, further studies are needed.

Reviewer 2 Report

The authors present a well written review article on a rare skin tumor, i.e., porocarcinoma.

I recommend the following modifications to the manuscript:

1) For incidence/epidemiology include the references:

--Incidence and trends of cutaneous adnexal tumors in the United States in 2000-2018: A population-based study. Gibbs DC, Yeung H, Blalock TW. J Am Acad Dermatol. 2022 May 4:S0190-9622(22)00779-4. doi: 10.1016/j.jaad.2022.04.052

--Head and Neck Porocarcinoma: SEER Analysis of Epidemiology and Survival. Scampa M, Merat R, Kalbermatten DF, Oranges CM. J Clin Med. 2022 Apr 14;11(8):2185. doi: 10.3390/jcm11082185

--Porocarcinoma: an epidemiological, clinical, and dermoscopic 20-year study. De Giorgi V, Silvestri F, Savarese I, Venturi F, Scarfì F, Trane L, Bellerba F, Zuccaro B, Maio V, Massi D, Gandini S. Int J Dermatol. 2022 Sep;61(9):1098-1105

2) For pathogenesis, please also include

--UV-induced local immunosuppression in the tumour microenvironment of eccrine porocarcinoma and poroma.

Puttonen M, Isola J, Ylinen O, Böhling T, Koljonen V, Sihto H. Sci Rep. 2022 Apr 1;12(1):5529. doi: 10.1038/s41598-022-09490-5

--High tumour mutational burden and EGFR/MAPK pathway activation are therapeutic targets in metastatic porocarcinoma.

Westphal D, Garzarolli M, Sergon M, Horak P, Hutter B, Becker JC, Wiegel M, Maczey E, Blum S, Grosche-Schlee S, Rütten A, Ugurel S, Stenzinger A, Glimm H, Aust D, Baretton G, Beissert S, Fröhling S, Redler S, Surowy H, Meier F. Br J Dermatol. 2021 Dec;185(6):1186-1199

3) For diagnosis please also include

--Comparison of Immunohistochemical Expression of Cytokeratin 19, c-KIT, BerEP4, GATA3, and NUTM1 Between Porocarcinoma and Squamous Cell Carcinoma. Goto K, Ishikawa M, Hamada K, Muramatsu K, Naka M, Honma K, Sugino T. Am J Dermatopathol. 2021 Nov 1;43(11):781-787

4) Please insert a new chapter on "Prognosis and risk factors for metastazation/recurrence/disease specific survival". Some aspects are already found in the text and can be transferred in the new chapter. Is the size of the tumor, the depth, the surgical safety margin or the sentinel node status relevant? Please also include

--Clear Cell Differentiation in Eccrine Porocarcinoma as a High-Risk Feature: Epidemiologic and Pathologic Features of Eccrine Porocarcinoma in a Single-Center Case Series. Koh M, Telang G, Fonseca A, Ghanian S, Walker J. Am J Dermatopathol. 2021 Sep 1;43(9):647-652

--Head and Neck Porocarcinoma: SEER Analysis of Epidemiology and Survival. Scampa M, Merat R, Kalbermatten DF, Oranges CM. J Clin Med. 2022 Apr 14;11(8):2185

--Porocarcinoma: an epidemiological, clinical, and dermoscopic 20-year study. De Giorgi V, Silvestri F, Savarese I, Venturi F, Scarfì F, Trane L, Bellerba F, Zuccaro B, Maio V, Massi D, Gandini S. Int J Dermatol. 2022 Sep;61(9):1098-1105

5) In the treatment chapter: For surgical treatment: please discuss also the potential role of the sentinel node excision. A possible reference here is

--Malignant Cutaneous Adnexal Tumors and Role of SLNB. Storino A, Drews RE, Tawa NE Jr. J Am Coll Surg. 2021 Jun;232(6):889-898.

6) In the treatment chapter, please include a separate subchapter on radiotherapy, both in the adjuvant setting als well as in the metastatic setting. Please include

--The role of postoperative radiotherapy in eccrine porocarcinoma: a multidisciplinary systematic review. Fionda B, Di Stefani A, Lancellotta V, Gentileschi S, Caretto AA, Casà C, Federico F, Rembielak A, Rossi E, Morganti AG, Schinzari G, Peris K, Tagliaferri L. Eur Rev Med Pharmacol Sci. 2022 Mar;26(5):1695-1700. 

Author Response

Reply to Reviewer #2:

The authors present a well written review article on a rare skin tumor, i.e., porocarcinoma.

I recommend the following modifications to the manuscript:

We appreciate the positive comments from Reviewer #2. We would like to reply to the comments point by point.

1) For incidence/epidemiology include the references:

--Incidence and trends of cutaneous adnexal tumors in the United States in 2000-2018: A population-based study. Gibbs DC, Yeung H, Blalock TW. J Am Acad Dermatol. 2022 May 4:S0190-9622(22)00779-4. doi: 10.1016/j.jaad.2022.04.052

--Head and Neck Porocarcinoma: SEER Analysis of Epidemiology and Survival. Scampa M, Merat R, Kalbermatten DF, Oranges CM. J Clin Med. 2022 Apr 14;11(8):2185. doi: 10.3390/jcm11082185

--Porocarcinoma: an epidemiological, clinical, and dermoscopic 20-year study. De Giorgi V, Silvestri F, Savarese I, Venturi F, Scarfì F, Trane L, Bellerba F, Zuccaro B, Maio V, Massi D, Gandini S. Int J Dermatol. 2022 Sep;61(9):1098-1105

Thank you for your comment. We have added citations for these references and information to the epidemiology section as follows:

Page 2, lines 54–57

Gibbs et al. analyzed the incidence rate and trend for cutaneous adnexal tumors in the United States between 2000 and 2018.[8] The age-adjusted incidence rate for PC was 0.045 per 100,000 person-years. They also mentioned that the incidence rate for PC did not change during the period 2000–2018.

Page 2, lines 79–82

Scampa et al. analyzed 563 cases of PC from the Surveillance, Epidemiology, and End Results (SEER) data.[15] In that analysis, the skin of the lower extremities and hips were the most common sites (190 cases: 33.7%), followed by the skin of the head and neck (30.6%) and the skin of the trunk (19.5%).

Page 2, lines 91–96

De Giorgi et al. reported that the affected sites of PC may depend on the sex, based on a study of 52 PC cases (31 male and 21 female). [17] Their study revealed that the most involved site was the head and neck (47%), which was more common in males than in females (34% males vs. 17% females). The lower extremities were the most common site among female cases (17% females vs. 9% males).

2) For pathogenesis, please also include

--UV-induced local immunosuppression in the tumour microenvironment of eccrine porocarcinoma and poroma.

Puttonen M, Isola J, Ylinen O, Böhling T, Koljonen V, Sihto H. Sci Rep. 2022 Apr 1;12(1):5529. doi: 10.1038/s41598-022-09490-5

--High tumour mutational burden and EGFR/MAPK pathway activation are therapeutic targets in metastatic porocarcinoma.

Westphal D, Garzarolli M, Sergon M, Horak P, Hutter B, Becker JC, Wiegel M, Maczey E, Blum S, Grosche-Schlee S, Rütten A, Ugurel S, Stenzinger A, Glimm H, Aust D, Baretton G, Beissert S, Fröhling S, Redler S, Surowy H, Meier F. Br J Dermatol. 2021 Dec;185(6):1186-1199

Thank you very much for your suggestions. The references you recommended were educational and interesting. We have added the information to the text (3. Pathogenesis section) as follows:

Page 3, lines 116–121

Puttonen et al. found a negative correlation between UV damage and tumor-infiltrating lymphocytes (TILs).[22] The results suggest that UV has a potentially immunosuppressive effect on PC microenvironments. On the other hand, in Merkel cell polyoma-virus-negative Merkel cell carcinoma, UV exposure correlates with increased TILs.[23] Further studies are needed on the relationship between UV exposure and TILs in PC.

Page 3, lines 145–148

Westphal et al. reported the overexpression of the genes EGFR, PAK1, and MAP2K1 (mitogen-activated protein kinase kinase 1; also known as MEK1) in PC tumors as assessed by WES, RNA sequencing (RNA-Seq), and comparative genomic hybridization analyses.[74] Based on this result, they treated the patient with anti-EGFR antibody.

3) For diagnosis please also include

--Comparison of Immunohistochemical Expression of Cytokeratin 19, c-KIT, BerEP4, GATA3, and NUTM1 Between Porocarcinoma and Squamous Cell Carcinoma. Goto K, Ishikawa M, Hamada K, Muramatsu K, Naka M, Honma K, Sugino T. Am J Dermatopathol. 2021 Nov 1;43(11):781-787

Thank you for your comment. The study by Goto et al. revealed that the immunohistochemical findings could provide a clue to differentiate SCC from porocarcinoma. We have cited this reference and added the information to the text (4-4. Immunohistochemistry).

Page 7, lines 249–254

Immunohistochemical stainings for cytokeratin 19, c-KIT, and BerEP4 are also useful when differentiating PC from SCC.[47] Goto et al. revealed that cytokeratin 19 was ex-pressed in 13 of 14 PC cases (92.9%) but was positive in only 9 of 22 SCC cases (40.9%). c-KIT was positive in 11 of 14 PC cases (78.6%), compared to only 3 of 22 SCC cases (13.6%). Moreover, BerEP4 expression also differed between PC and SCC (57.1% vs. 9.1%, respectively).

4) Please insert a new chapter on "Prognosis and risk factors for metastazation/recurrence/disease specific survival". Some aspects are already found in the text and can be transferred in the new chapter. Is the size of the tumor, the depth, the surgical safety margin or the sentinel node status relevant? Please also include

--Clear Cell Differentiation in Eccrine Porocarcinoma as a High-Risk Feature: Epidemiologic and Pathologic Features of Eccrine Porocarcinoma in a Single-Center Case Series. Koh M, Telang G, Fonseca A, Ghanian S, Walker J. Am J Dermatopathol. 2021 Sep 1;43(9):647-652

--Head and Neck Porocarcinoma: SEER Analysis of Epidemiology and Survival. Scampa M, Merat R, Kalbermatten DF, Oranges CM. J Clin Med. 2022 Apr 14;11(8):2185

--Porocarcinoma: an epidemiological, clinical, and dermoscopic 20-year study. De Giorgi V, Silvestri F, Savarese I, Venturi F, Scarfì F, Trane L, Bellerba F, Zuccaro B, Maio V, Massi D, Gandini S. Int J Dermatol. 2022 Sep;61(9):1098-1105

Thank you for your comment. We have inserted a new chapter: “5. Prognosis and risk factors for metastasis/recurrence/disease-specific survival”. In accordance with your suggestions, we have added the information related to the prognosis and risk factors to the text as follows.

Pages 7–8, lines 272–314

  1. Prognosis and risk factors for metastasis/recurrence/disease-specific survival

The prognosis for PC seemed to be generally good in early-stage cases, which are almost always treated with surgical resection. In patients with malignant cutaneous adnexal tumors (MCATs), the overall 5-year survival rate was 96% for those with stage I or II.[50] There has been no consensus on the surgical margin for wide local resection (WLE). In general, the average margin is 2–3 cm for surgical resection.[51] WLE has been found to have a local recurrence rate of 20% even when clear margins are confirmed at primary resection. Therefore, Mohs micrographic surgery (MMS) has been increasingly used, with reports of less local recurrence than with WLE.

Histopathological analysis is important when determining tumor invasion and for the prediction of metastasis/recurrence. Robert A et al. reported that the tumor borders of PC can be divided into “infiltrative”, “pagetoid”, or “pushing”.[5] “Infiltrative” is characterized by an ill-defined lower margin of malignant clusters infiltrating the dermis. “Pagetoid” is identified by the intraepidermal spread of tumor cells mimicking Paget’s disease. “Pushing” PC lesions often have polypoid tumors with distinct dermal limits.[38] Robert A et al. also reported that some histopathological characteristics reflect poor prognosis, known as “poor-prognosis histopathological signs”.[5] The signs are the presence of lympho-vascular invasion, more than 14 mitoses per high-power field, and tumoral involvement deeper than 7 mm.[14] Koh et al. investigated epidemiological and pathological features of 52 PC cases and determined several high-risk features.[52] Tumor depth was associated with both increased age and clear cell differentiation, suggesting it as a high-risk factor with poor prognosis.

As for the recurrence rates, a retrospective study revealed that 6 of 52 cases (11.5%) showed recurrence of PC tumors within 5 years.[17] Since the reported recurrence rates vary from study to study, Belin et al. proposed an algorithm using prognostic histological factors that could guide surgical procedures.[38] First, they classified biopsy specimens into the three subtypes of infiltrative, pushing, and pagetoid as described above. If the resected sample was pushing PC, no additional resection was performed, but if it was infiltrative or pagetoid, they recommended additional MMS.

Scampa et al. reported the 5-year overall survival (OS) as 74.8% based on an analysis of 563 PC cases.[15] They also mentioned that the prognosis for PC varies greatly between early and advanced stages. In patients with advanced/metastatic PC, the prognosis is poor. A 2017 meta-analysis of 453 cases reported that 110 cases (31%) presented metastasis, and the most common metastatic organ was the nearby lymph node (58.5%), followed by the lungs (12.8%).[11] Song et al. mentioned that in the presence of lymph node metastasis, the mortality rate is as high as 67%, and the probability of distant metastasis is about 11%–12%.[51] Robson et al. found that patients with lymph node metastases had 1-year and 3-year overall survival (OS) of 88.9% and 39.5%, respectively.[5] No studies have directly compared OS for metastatic SCC to that for metastatic PC. Hillen et al. analyzed 190 cases of advanced SCC[53] and found a 1-year OS of 84% and a 3-year OS of 47% for metastatic SCC. Based on these results, the overall survival rate for metastatic PC may be worse than that for metastatic SCC at 3 years. Concerning the prognostic difference between SCC and PC, further studies are needed.

5) In the treatment chapter: For surgical treatment: please discuss also the potential role of the sentinel node excision. A possible reference here is

--Malignant Cutaneous Adnexal Tumors and Role of SLNB. Storino A, Drews RE, Tawa NE Jr. J Am Coll Surg. 2021 Jun;232(6):889-898.

In accordance with your comment, we have added a citation for this reference and added the information on the role of SLNB to the text (6-1. Surgical treatments).

Page 8, lines 324–330

Few reports have addressed sentinel lymph node biopsy (SLNB) in MCATs, and no definite conclusions have been reached. Storino et al. investigated 41 patients diagnosed with MCATs.[50] Of the 25 patients who underwent SLNB, only one had lymph node metastasis, suggesting that SLNB may not be necessary in early-stage cases of MCATs. However, if the SLNB was positive, metastatic lesions could be detected early and systemic treatment could be initiated. SLNB could be considered depending on the patient's condition and the presence of advanced disease risk factors.

6) In the treatment chapter, please include a separate subchapter on radiotherapy, both in the adjuvant setting as well as in the metastatic setting. Please include

--The role of postoperative radiotherapy in eccrine porocarcinoma: a multidisciplinary systematic review. Fionda B, Di Stefani A, Lancellotta V, Gentileschi S, Caretto AA, Casà C, Federico F, Rembielak A, Rossi E, Morganti AG, Schinzari G, Peris K, Tagliaferri L. Eur Rev Med Pharmacol Sci. 2022 Mar;26(5):1695-1700.

Thank you very much for your helpful comments. In accordance with the comments, we have added the subchapter “radiation therapy” to the treatment chapter. We have mentioned the role of radiation therapy for porocarcinoma based on your recommended reference. There have been no established criteria for radiotherapy against porocarcinoma, and we found only one case report in which primary radiotherapy was applied for a metastatic porocarcinoma patient. Concerning the role of radiotherapy in an adjuvant setting, we have cited the reference and added the information to the text as follows.

Pages 8–9, lines 332–340

6-2. Radiotherapy

There are no clear criteria for radiation therapy (RT) for PC.[55] There has been only one case in which primary RT was used to treat PC lesions that were not suitable for surgery.[54] Reports on the use of adjuvant RT after surgery are also limited. Adjuvant RT has generally been considered helpful for lymph node metastases, perineural invasion, positive resection margins, high-grade tissue, multiple lesions, and recurrent disease. It is often used in combination with chemotherapy.[56] Fionda et al. published a systematic review of adjuvant RT against PC,[55] suggesting that postoperative adjuvant RT may be effective in cases with positive or close margins and unfavorable histology.

Round 2

Reviewer 2 Report

Thank you for performing the revision.

I habe just a few minor comments:

line 70: "four" reports instead of three

line 95: "a" metaanalysis, not the

line 299/300: stage I and stage II is mentioned, this staging system should be explained (reference, how are the different stages defined)

Author Response

Reply to Reviewer #2:

Thank you for performing the revision.

I habe just a few minor comments:

Thank you very much for reviewing our manuscript and offering valuable advice.

We have addressed your comments with point-by-point responses and revised the manuscript accordingly.

1)line 70: "four" reports instead of three

Thank you for your comment. We have revised the manuscript accordingly.

Page 2, lines 62–63

The incidence rate of PC is at least 10 times higher in England than in the locations of the other four reports above.

2)line 95: "a" metaanalysis, not the

Thank you for your comment. We have changed the texts accordingly.

Page 2, lines 87–88

Salih et al. performed a meta-analysis of PC.[11]

3)line 299/300: stage I and stage II is mentioned, this staging system should be explained (reference, how are the different stages defined)

Thank you for your helpful comment. In accordance with your suggestions, we have added the information on the staging system for MCATs to the text.

Pages 7, lines 274–276

The staging for MCATs has been defined by the AJCC TNM staging manual, which is the same staging system for SCC.[50]